# NodeCwR: Node Classification with Reject Option

## Abstract

Graph attention networks (GAT) have been state-of-the-art GNN architecture used as the backbone for various graph learning problems. One of the key tasks in graph learning is node classification. While several works cover multiple aspects of node classification, there has yet to be an attempt to understand the behavior of GAT models for node classification with a reject option. This paper proposes a new approach called Node-CwR, which models node classification with a reject option using GAT. We offer both cost-based and coverage-based models to include the reject option in the node classification task. Cost-based models find the optimal classifier for a given cost of rejection value. Such models are trained by minimizing the rejection and misclassification rates on unrejected samples. Coverage-based methods take coverage as input and find the optimal model for a given coverage rate. We empirically evaluate our approaches on three benchmark datasets and show their effectiveness in learning efficient reject option models for node classification tasks. We observe that, in general, cost-based methods outperform coverage-based models for reject option. Additionally, our results include robust learning of node classifiers using label smoothing in the presence of label noise. We observe that label smoothing works well to handle label noise in cost-based models, while it works adversely in coverage-based models.

## 1 Introduction

The recent decade has witnessed a surge of interest in using graph neural networks (GNNs) in various domains such as computer vision Satorras & Estrach (2018), natural language processing Schlichtkrull et al. (2017), and bioinformatics Xia & Ku (2021), to name a few. GNNs (Kipf & Welling, 2017) capture structural aspects of the data in the form of nodes and edges to perform any prediction task. It learns node, edge, and graph-level embeddings to get high-dimensional features using the message-passing mechanism in the GNN layer. Recently, graph attention networks (GAT) (Veličković et al., 2018) have been shown to work well for node-level tasks as they leverage the attention mechanism to learn the importance of neighboring nodes and scale their features accordingly before performing a linear combination of the nodes. The most common node-level task is a node classification task, which uses graph-structured data to classify nodes.

Deep graph learning is also used in the field of responsible AI in high-risk applications such as legal judgment prediction Dong & Niu (2021), disease prediction Sun et al. (2021), financial fraud prediction Xu et al. (2021), etc., with the high cost of incorrect predictions. It is challenging to overcome such issues while using standard GNN architectures. To deal with such scenarios of high-risk applications, the model shouldn't make a prediction and further examine the features instead of making a wrong prediction with potential consequences. Can we allow GNN models to abstain in case of insufficient confidence for prediction?

Reject option classifiers have an additional option of refraining from deciding when in a dilemma. The attraction of reject option classification is evident in applications where one can afford partial domain coverage and where extremely low risk is a must. Still, it is not achievable in standard classification frameworks. Consider the case of the diagnosis of a patient for a specific disease. In case of confusion, the physician might choose not to risk misdiagnosing the patient. She might instead recommend further medical tests to the patient or refer her to an appropriate specialist. The primary response in these cases is to "reject" the example. Typically, there is a cost involved in

rejecting an example. However, that cost is much smaller compared to misclassification. The goal of integrating a reject option in the classification model is to allow it to abstain from giving a prediction when it is highly uncertain to make a decision. These models are usually given the option not to make a prediction based on the following two ideas: i) Coverage-based model: Coverage is defined as the ratio of samples that the model does not reject. For a given coverage, the model finds the best examples that can give the best performance. The coverage can further be varied post-training by our choice of threshold. SelectiveNet (El-Yaniv et al., 2010; Geifman & El-Yaniv, 2019) is a recent work that proposes a coverage based deep neural network (DNN) approach for the reject option classifier. ii) Cost-based model: These models include the cost of rejection and misclassification. The cost of rejection can vary based on the application. The model optimizes according to the cost to minimize the number of examples rejected and misclassified. The loss in such approaches is $l_{0d1}$ (Chow, 1970). However, in practice, surrogates of $l_{0d1}$ (Bartlett & Wegkamp, 2008; Grandvalet et al., 2009; Manwani et al., 2015; Shah & Manwani, 2019; Cortes et al., 2016) are used. DNN-based reject option classifier are proposed in Kalra et al. (2021); Ni et al. (2019); Charoenphakdee et al. (2021); Cao et al. (2022).

However, the reject option for the node classification problem using GNN has not been attempted. Motivated by the abovementioned observations, we propose rejection-based GAT networks for the node classification tasks, which we call NodeCwR. We consider an integrated reject option using cost-based (NodeCwR-Cost) and coverage-based (NodeCwR-Cov) models. We can use NodeCwR-Cost when the cost of rejecting an example and NodeCwR-Cov when the ratio of examples to predict is specified. We also explore label smoothing to see if NodeCwR becomes robust when there is label noise. Our contributions are as follows: **i.)** To the best of our knowledge, we are the first to learn node embeddings using abstention-based GAT architecture. **ii.)** We extend and generalize GAT to train for node features with cost-based and coverage-based abstention models. **iii.)** Finally, we perform an empirical study to evaluate our models on popular benchmark datasets for node classification tasks. Our experimental results show that NodeCwR-Cost performs better than NodeCwR-Cov in general. **iv.)** We also show experimentally that NodeCwR-Cost with label smoothing becomes robust against label noise.

## 2 RELATED WORK

### 2.1 REJECT OPTION CLASSIFICATION

There are two broad categories of approaches for reject option classifiers: coverage-based and cost-based. Coverage-based learn based on optimizing risk-coverage trade-offs. SelectiveNet is a coverage-based method proposed for learning with abstention (El-Yaniv et al., 2010; Geifman & El-Yaniv, 2019). SelectiveNet is a deep neural network architecture that optimizes prediction and selection functions to model a selective predictor. As this approach does not consider rejection cost $d$ in their objective function, it can avoid rejecting hazardous examples. This, in particular, becomes a severe issue in high-risk situations (e.g., healthcare systems, etc.). Cost-based approaches assume that the reject option involves a cost $d$. Generalized hinge SVM (Bartlett & Wegkamp, 2008), double hinge SVM (Grandvalet et al., 2009), double ramp SVM (Manwani et al., 2015), SDR-SVM (Shah & Manwani, 2019), max-hinge SVM and plus-hinge SVM (Cortes et al., 2016) etc. are some variants of support vector machine (SVM) for reject option classifiers. Nonlinear classifiers in these approaches are learned using kernel functions. In (Kalra et al., 2021), the authors propose a deep neural network-based reject option classifier for two classes that learn instance-dependent rejection functions. In Ramaswamy et al. (2018), multiclass extensions of the hinge-loss with a confidence threshold are considered for reject option classification. Ni et al. (2019) prove calibration results for various confidence-based smooth losses for multiclass reject option classification. Charoenphakdee et al. (2021) prove that $K$-class reject option classification can be broken down into $K$ binary cost-sensitive classification problems. They subsequently propose a family of surrogates, the ensembles of arbitrary binary classification losses. Cao et al. (2022) propose a general recipe to convert any multiclass loss function to accommodate the reject option, calibrated to loss $l_{0d1}$. They treat rejection as another class.

## 2.2 Node Classification

Node classification is a fundamental task related to machine learning for graphs and network analysis. GNN methods can be broadly classified into three categories that perform node classification as the primary task. The first set of models introduced convolution-based GNN architectures by extending original CNNs to graphs Scarselli et al. (2008), Defferrard et al. (2016), Hamilton et al. (2017), Kipf & Welling (2017) Bresson & Laurent (2017). Secondly, proposed attention and gating mechanism-based architectures using anisotropic operations on graphs Veličković et al. (2018). The third category focuses on the theoretical limitations of previous types Xu et al. (2018), Morris et al. (2019), Maron et al. (2019), Chen et al. (2019).

## 2.3 Label Smoothing

Label smoothing uses a positively weighted combination of hard training labels and uniformly distributed soft labels. It was first proposed in Szegedy et al. (2016) for model regularization in the context of CNNs. When there are noisy labels, label smoothing is a robust learning approach (Lukasik et al., 2020). Wei et al. (2021) introduce the concept of negative label smoothing (NLS) in which soft labels for some classes can be negative. They show that NLS is more effective than simple label smoothing in case of a high noise rate.

# 3 Method

GAT focuses on learning effective and efficient representations of nodes to perform any downstream task. Let $\mathcal{X}$ be the instance space and $\mathcal{Y} \in \{1, \ldots, K\}$ be the label space. We represent the embedding space learned using GAT by $\mathcal{H}$. Thus, GAT learns a mapping $GAT : \mathcal{X} \to \mathcal{H}$. GAT treats each instance as a node and learns embedding for each node.

## 3.1 NodeCwR-Cov: Coverage Based Node Classifier With Rejection

NodeCwR-Cov uses coverage-based logic to learn node classifiers with a reject option. We use similar ideas to SelectiveNet (Geifman & El-Yaniv, 2019) to learn the coverage-based rejection function. Figure 1 shows the architecture of NodeCwR-Cov. Node representations are learned using first GAT layer and given as input to second GAT layer which follows softmax layer. The second GAT layer and softmax layer combined learn mapping $\mathbf{f} : \mathcal{H} \to \Delta_{K-1}$ where $\Delta_{K-1}$ is $K$-dimensional simplex. Function $\mathbf{f}$ is used to predict the class of a node. There are two more fully connected layers after the softmax layer (having 512 nodes and one node) to model the selection function $g : \mathcal{H} \to \{0, 1\}$. Selection function $g$ decides whether to predict a given example or not. Selection function $g(\mathbf{h})$ is a single neuron with a sigmoid activation. At the beginning, a threshold of 0.5 is set for the selection function, which means $\mathbf{f}(\mathbf{h})$ predicts $\mathbf{h}$ if and only if $g(\mathbf{h}) \geq 0.5$. The auxiliary prediction head implements the prediction task $a(\mathbf{h})$ without the need for coverage to get a better representation of examples with low confidence scores, which are usually ignored by the prediction head. This head is only used for training purposes. We use cross-entropy loss $l_{ce}$ to capture the error made by the prediction function $\mathbf{f}(\mathbf{h})$. The empirical risk of the model is captured as follows.

$$r(\mathbf{f}, g|S_n) = \frac{\frac{1}{n} \sum_{i=1}^{n} l(f(\mathbf{h}_i), y_i)g(\mathbf{h}_i)}{\phi(g|S_n)}$$

where $\phi(g|S_n)$ is empirical coverage computed as $\phi(g|S_n) = \frac{1}{n} \sum_{i=1}^{n} g(\mathbf{h}_i)$. An optimal selective model could be trained by optimizing the selective risk given constant coverage. We use the following error function to optimize $\mathbf{f}(.)$ and $g(.)$.

$$E(\mathbf{f}, g) = r(\mathbf{f}, g|S_n) + \lambda \Psi(c - \phi(g|S_n))$$

where $\Psi(a) = \max(0, a)^2$ is a quadratic penalty function, $c$ is the target coverage, and $\lambda$ controls the importance of coverage constraint. The loss function used at the auxiliary head is standard cross entropy loss $l_{ce}$ without any coverage constraint. Thus, the empirical risk function corresponding to the auxiliary head is $E(\mathbf{f}) = 1/n \sum_{i=1}^{n} l_{ce}(\mathbf{f}(\mathbf{h}_i), y_i)$. The final error function of NodeCwR-Cov is a convex combination of $E(f, g)$ and $E(\mathbf{f})$ as follows, $E = \alpha E(\mathbf{f}, g) + (1 - \alpha)E(\mathbf{f})$, where $\alpha \in (0, 1)$. When the data is trained over a training set using a coverage constraint, this constraint

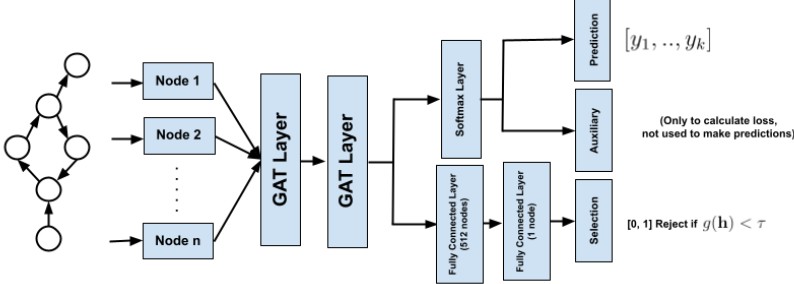

Figure 1: Architecture of NodeCwR-Cov: Coverage based node classifier with rejection

is violated on the test set. The constraint requires the true coverage $\phi(g)$ to be larger than the given coverage constraint $c$, which is usually violated. To get the optimal actual coverage, we calibrate the threshold $\tau$ to select the example in $g(h')$ using this validation set, which results in coverage as close as possible to target coverage.

## 3.2 NODECWR-COST: COST BASED NODE CLASSIFIER WITH REJECTION

In the cost-based method, the cost of rejection $d$ is pre-specified. The architecture of NodeCwR-Cost is presented in Figure 2. The first block in NodeCwR-Cost consists of two GAT layers. The output of the second GAT layer is fed to a softmax layer with $(K + 1)$ nodes. Note that we assume rejection as the $(K + 1)^{th}$ class here. The second GAT layer and softmax layer combined learn prediction function $f : \mathcal{H} \to \Delta_K$ where $\Delta_K$ is $(K + 1)$-dimensional simplex. Note that $(K + 1)^{th}$ output corresponds to the reject option in this architecture. Let $\mathbf{e}_j$ denote $K + 1$-dimensional vector such that its $j^{th}$ element is one and other elements are zero. Note that $(K + 1)^{th}$ element never becomes one as we do not get a rejection label in the training data. We use the following variant of cross-entropy loss, which also incorporates the cost of rejection (Cao et al., 2022).

$$l_{ce}^d(f(\mathbf{h}), \mathbf{e}_y, \mathbf{e}_{K+1}) = l_{ce}(f(\mathbf{h}), \mathbf{e}_y) + (1 - d)l_{ce}(f(\mathbf{h}), \mathbf{e}_{K+1})$$
$$= -\log f_y(\mathbf{h}) - (1 - d)\log f_{K+1}(\mathbf{h})$$

Here $f_{K+1}(\mathbf{h})$ is the output corresponding to the reject option, and $f_y(\mathbf{h})$ is the output related to the actual class. For very small values of $d$, the model focuses more on maximizing $f_{K+1}(\mathbf{h})$ to prefer rejection over misclassification. Note that loss $l_{ce}^d$ is shown to be consistent with the $l_{0d1}$ loss (Cao et al., 2022). For $d = 1$, the loss $l_{ce}^d$ becomes the same as standard cross entropy loss $l_{ce}$. We represent smooth label of $j \in \{1, \ldots, K + 1\}$ with a $K + 1$-dimensional vector $\mathbf{e}_j^{LS}$ as $\mathbf{e}_j^{LS} = (1 - \epsilon)\mathbf{e}_j + \frac{\epsilon}{K+1}\mathbf{1}$. Here $\epsilon \in (0, 1)$ is the smoothing parameter, $\mathbf{1}$ is a $K + 1$-dimension vector whose all elements are one. $l_{ce}^d$ loss with smooth label is as follows.

$$l_{ce}^d(f(\mathbf{h}), \mathbf{e}_y^{LS}, \mathbf{e}_{K+1}^{LS}) = l_{ce}(f(\mathbf{h}), \mathbf{e}_y^{LS}) + (1 - d)l_{ce}(f(\mathbf{h}), \mathbf{e}_{K+1}^{LS})$$

$$= (1 - \epsilon)l_{ce}(f(\mathbf{h}), \mathbf{e}_y) + \frac{\epsilon}{K + 1}l_{ce}(f(\mathbf{h}), \mathbf{1})$$

$$+ (1 - d)\left[(1 - \epsilon)l_{ce}(f(\mathbf{h}), \mathbf{e}_{K+1}) + \frac{\epsilon}{K + 1}l_{ce}(f(\mathbf{h}), \mathbf{1})\right]$$

$$= (1 - \epsilon)l_{ce}^d(f(\mathbf{h}), \mathbf{e}_y, \mathbf{e}_{K+1}) + \frac{(2 - d)\epsilon}{K + 1}l_{ce}(f(\mathbf{h}), \mathbf{1})$$

Thus, we also do smoothing over $(K + 1)^{th}$ class which is rejection. We observe that this way of smoothing gives better performance in experiments. The extra term $\frac{(2-d)\epsilon}{K+1}l_{ce}(f(\mathbf{h}), \mathbf{1})$ acts as a regularizer which penalizes the model for going away from the uniform distribution $\frac{\epsilon}{K+1}\mathbf{1}$.

## 4 EXPERIMENTAL STUDY

This section describes the implementation details of NodeCwR-Cost and NodeCwR-Cov, their performance evaluation, and their comparison. We also present results with label noise.

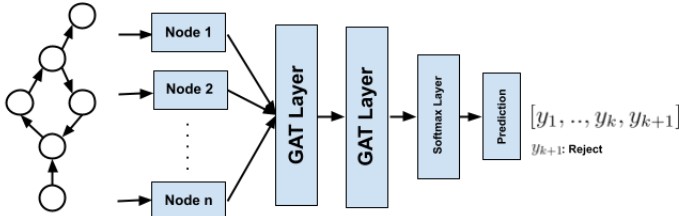

Figure 2: Architecture of NodeCwR-Cost: Cost based node classifier with rejection

## 4.1 DATATSETS USED

We evaluated our model on three standard citation network datasets, Cora, Citeseer, and Pubmed (Sen et al., 2008). In these data sets, each document is represented by a node, and the class label represents the category of the document. Undirected edges represent citations. For training, we use 20 nodes per class. Thus, the number of nodes for training varies for each data set depending on the number of classes. We use 500 nodes for validation and 1000 for testing on all the data sets.

## 4.2 GAT IMPLEMENTATION DETAILS

We used GAT as the base architecture for node classification and closely followed the experimental setup mentioned in Veličković et al. (2018). We modified the GAT implementation available by Antognini (2021) to implement our approach. We first applied dropout Srivastava et al. (2014) on node features with $p = 0.6$. These node features, along with the adjacency matrix, are passed through a GAT Layer having eight attention heads, where each head produces eight features per node. We used LeakyReLU as the activation function inside the GAT Layer with $\alpha = 0.2$. These outputs are concatenated for the first layer (64 features per node). Another dropout layer with the same probability follows this. The dropout layer's output is passed through the final GAT layer with a single attention head, which takes 64 features per node and outputs $k$ features per node, where $k$ is the number of classes. It uses ELU Clevert et al. (2015) activation function. The network output is passed through a softmax layer to get class posterior probabilities.

## 4.3 NODECWR-COV IMPLEMENTATION DETAILS

We use the GAT architecture noted above to integrate the coverage-based reject option into the model as mentioned in Geifman & El-Yaniv (2019). The output from the softmax layer is separately given to both prediction head $f$ and auxiliary head $h$. The output from the second GAT Layer is passed through a Fully Connected Hidden Layer with 512 nodes. This is passed through Batch Normalization Ioffe & Szegedy (2015), ReLU, and a Fully Connected Output Layer with one node. This is passed through Sigmoid Activation to get a selection score $[0, 1]$. The Prediction head and Selection head are concatenated together, and the selective loss is performed on this output. We set $\lambda = 32$ as the constraint on coverage to calculate this loss. Cross Entropy Loss is performed on the output of the Auxiliary head but is not used for making predictions. A convex combination of these two loss values with $\alpha^l = 0.5$ is used to optimize the model. We trained the model with early stopping with patience of 100 epochs. Error on the validation set is used to implement the stopping condition. We observed that the model trains for approximately 1800 epochs to train the NodeCwR-Cov model. Once the model is trained, the number of covered examples on the test data set when $\tau = 0.5$ will vary highly because the model is not used for the test data. However, since we have the selection scores of each node, we sort them and select a $\tau$ value that matches the coverage we expect.

## 4.4 NODECWR-COST IMPLEMENTATION DETAILS

We also trained NodeCwR-Cost by taking the output of the GAT network. In the cost-based approach, we treat the reject option as an integrated class in the model. Hence, for a $k$ class classification problem, we change the model architecture from giving $k$ outputs to $k + 1$ outputs. In this

model, we have to perform CwR Loss for optimization. From the output, every node that gets $k + 1$ as the output will be rejected. Here also, we used early stopping with a patience of 100 epochs. Model trains for approximately 1000 epochs for NodeCwR-Cost.

## 4.5 RESULTS ON NODECWR-COV

We repeat each experiment 10 times with random initialization and report the average and standard deviation. We see that the performance of NodeCwR-Cov decreases with label smoothing except for the Citeseer dataset. For the Citeseer dataset, for $d$=0.1,0.2,0.8,0.9 $LS = \epsilon$ dominates $LS = 0$. However, the gain is marginal.

| | Cora | | Pubmed | | Citeseer | |
|---|---|---|---|---|---|---|
| **Cov** | **Acc(LS=0)** | **Acc(LS=0.5)** | **Acc(LS=0)** | **Acc(LS=0.5)** | **Acc(LS=0)** | **Acc(LS=0.5)** |
| 0.1 | **97.8 ± 0.84** | 96.13 ± 2.42 | **93.4 ± 2.61** | 87.2 ± 5.36 | 89 ± 4.42 | **89.2 ± 4.6** |
| 0.2 | **97.1 ± 0.77** | 94.75 ± 1.86 | 80.4 ± 10.6 | **80.6 ± 12** | 87.5 ± 2.62 | **88.1 ± 1.92** |
| 0.3 | **96.43 ± 0.85** | 93.33 ± 1.43 | **85.5 ± 3.49** | 82.1 ± 3.81 | **85.6 ± 2.5** | 84.7 ± 3.42 |
| 0.4 | **95.62 ± 1.16** | 92.38 ± 0.77 | **85.3 ± 1.74** | 82.2 ± 2.23 | **85.2 ± 1.46** | 82.8 ± 3.44 |
| 0.5 | **93.96 ± 1.45** | 92.18 ± 1.1 | **83.2 ± 3.34** | 79.9 ± 2.47 | **81.3 ± 2.19** | 76.6 ± 6.06 |
| 0.6 | **92.65 ± 0.5** | 91.29 ± 1.32 | **82.6 ± 1.48** | 82.5 ± 1.87 | **79.6 ± 2.43** | 78.4 ± 4 |
| 0.7 | **91.29 ± 0.45** | 90.25 ± 1.7 | **79.8 ± 2.46** | 79.5 ± 0.98 | **75.9 ± 2.86** | 74.8 ± 2.12 |
| 0.8 | **89.12 ± 0.8** | 88.5 ± 1.43 | 80.9 ± 1.24 | **81 ± 0.44** | 72.8 ± 1.05 | **73.2 ± 2.29** |
| 0.9 | **86.65 ± 0.7** | 85.81 ± 1.03 | **79.7 ± 0.59** | 79 ± 1.17 | 72 ± 0.69 | **74.3 ± 0.41** |

Table 1: Accuracy and Coverage of GAT + CwR at different cost of rejection.

Although we can calibrate the threshold to cover any number of examples irrespective of the training coverage, it is preferred to train the model on similar coverage rates and then calibrate it to our preferred coverage to get the best results.

## 4.6 RESULTS ON NODECWR-COST

We repeat each experiment 10 times with random initialization and report the average and standard deviation. As the $d$ increases, the rejection rate decreases, increasing coverage. As the coverage increases, more misclassifications happen, which decreases the accuracy. We observe this trend even with label smoothing. This behavior is expected from a cost-based rejection model.

We observe that as the $d$ increases, 0-d-1 risk increases initially, and it starts dropping after a certain value of $d$. These values of $d$ are called crossover values. These crossover values are 0.55 for Cora, 0.45 for Pubmed, and 0.65 for Citeseer. For smaller values of $d$, when coverage increases, the misclassification rate also increases, which causes an increase in 0-d-1 risk. But, as $d$ increases, the difference between misclassification cost (which is one) and rejection cost ($d$) reduces.

We observe an interesting behavior of 0-d-1 risk, which favors $LS = 0$ (no label smoothing) for smaller $d$ values and favors $LS = \epsilon$ (label smoothing) for higher values of $d$. This happens due to the following reasons. Label smoothing allows each label to exist with non-zero probability, for example. This creates more confusion areas for the classifier. This causes $LS = \epsilon$ to make more rejections for smaller $d$ values than $LS = 0$. This makes $LS = \epsilon$ incur higher $0 - d - 1$ risk than $LS = 0$ for smaller $d$ values. For higher values of $d$, the rejection rate decreases for $LS = \epsilon$ and $LS = 0$. As $d$ increases, coverage increases, and the NodeCwR-Cost approach focuses more on improving classification accuracy. At this point, the true power of $LS = \epsilon$ comes into the picture due to its regularization effect. Thus, NodeCwR-Cost with $LS = \epsilon$ starts performing better for higher $d$ values than $LS = 0$.

While we can calibrate the number of examples we want to predict in selection-based models, we can only choose a cost and let the model reject any examples in cost-based models. Hence, we plotted the accuracy with respect to the coverage of both these models for better comparison.

| | $d$ | LS=0 | | | LS=0.5 | | |
| --- | --- | --- | --- | --- | --- | --- | --- |
| | | Acc. on Unrejected | Coverage | 0-d-1 Risk | Acc. on Unrejected | Coverage | 0-d-1 Risk |
| **Cora** | 0.1 | $97.5 \pm 0.48$ | $\mathbf{12.1 \pm 0.95}$ | $\mathbf{0.091 \pm 0.001}$ | $98.5 \pm 1.52$ | $5.5 \pm 0.97$ | $0.095 \pm 0.001$ |
| | 0.2 | $97.5 \pm 0.47$ | $\mathbf{17.1 \pm 1.6}$ | $\mathbf{0.17 \pm 0.003}$ | $98.1 \pm 0.68$ | $10.7 \pm 1.14$ | $0.181 \pm 0.003$ |
| | 0.3 | $97.5 \pm 0.68$ | $\mathbf{23.8 \pm 1.75}$ | $\mathbf{0.235 \pm 0.005}$ | $97.6 \pm 0.38$ | $18.8 \pm 2.41$ | $0.248 \pm 0.007$ |
| | 0.4 | $96.6 \pm 0.62$ | $\mathbf{32.2 \pm 0.91}$ | $\mathbf{0.282 \pm 0.004}$ | $98.1 \pm 0.21$ | $30.9 \pm 2.09$ | $0.283 \pm 0.008$ |
| | 0.5 | $95.8 \pm 0.23$ | $42.6 \pm 1.88$ | $0.305 \pm 0.009$ | $96.6 \pm 0.39$ | $\mathbf{45.9 \pm 1.53}$ | $\mathbf{0.286 \pm 0.007}$ |
| | 0.6 | $95 \pm 0.49$ | $53.1 \pm 1.07$ | $0.308 \pm 0.006$ | $94.1 \pm 0.55$ | $\mathbf{61 \pm 0.4}$ | $\mathbf{0.27 \pm 0.003}$ |
| | 0.7 | $92.8 \pm 0.55$ | $66.3 \pm 1.81$ | $0.283 \pm 0.009$ | $90.2 \pm 0.52$ | $\mathbf{80.4 \pm 0.82}$ | $\mathbf{0.216 \pm 0.006}$ |
| | 0.8 | $90.1 \pm 0.51$ | $83.3 \pm 0.68$ | $0.216 \pm 0.008$ | $84.6 \pm 0.42$ | $\mathbf{98 \pm 0.34}$ | $\mathbf{0.167 \pm 0.003}$ |
| | 0.85 | $87.2 \pm 0.78$ | $90.5 \pm 1.11$ | $0.196 \pm 0.003$ | $83.7 \pm 0.7$ | $\mathbf{100 \pm 0.04}$ | $\mathbf{0.163 \pm 0.007}$ |
| **Pubmed** | 0.1 | $97.4 \pm 1.86$ | $\mathbf{8.6 \pm 3.12}$ | $\mathbf{0.094 \pm 0.001}$ | $100$ | $0.6 \pm 0.25$ | $0.099$ |
| | 0.2 | $97.7 \pm 2.49$ | $\mathbf{12.9 \pm 4.52}$ | $\mathbf{0.178 \pm 0.005}$ | $98.8 \pm 2.63$ | $1.5 \pm 0.47$ | $0.197 \pm 0.001$ |
| | 0.3 | $93.5 \pm 1.2$ | $\mathbf{21.9 \pm 2.77}$ | $\mathbf{0.249 \pm 0.007}$ | $95.8 \pm 1.62$ | $7.9 \pm 3.08$ | $0.28 \pm 0.008$ |
| | 0.4 | $92.4 \pm 0.66$ | $\mathbf{27 \pm 5.33}$ | $\mathbf{0.313 \pm 0.016}$ | $93.1 \pm 0.95$ | $24.2 \pm 1.22$ | $0.32 \pm 0.005$ |
| | 0.5 | $88.9 \pm 0.92$ | $49.3 \pm 5.48$ | $\mathbf{0.309 \pm 0.018}$ | $87.7 \pm 0.94$ | $\mathbf{50.2 \pm 3.36}$ | $0.311 \pm 0.01$ |
| | 0.6 | $84.6 \pm 0.87$ | $67.8 \pm 4.81$ | $0.298 \pm 0.019$ | $81.2 \pm 0.29$ | $\mathbf{84.4 \pm 3.5}$ | $\mathbf{0.252 \pm 0.015}$ |
| **Citeseer** | 0.1 | - | $\mathbf{0}$ | $\mathbf{0.1}$ | - | $\mathbf{0}$ | $\mathbf{0.1}$ |
| | 0.2 | $100$ | $\mathbf{0.3 \pm 0.17}$ | $\mathbf{0.199}$ | - | $0$ | $0.2$ |
| | 0.3 | $99.2 \pm 1.72$ | $\mathbf{1.2 \pm 0.85}$ | $\mathbf{0.297 \pm 0.002}$ | - | $0$ | $0.3$ |
| | 0.4 | $93.7 \pm 2.26$ | $\mathbf{4.3 \pm 1.18}$ | $\mathbf{0.386 \pm 0.004}$ | - | $0$ | $0.4$ |
| | 0.5 | $91.6 \pm 2.23$ | $\mathbf{9.7 \pm 1.7}$ | $\mathbf{0.46 \pm 0.005}$ | $95.9 \pm 2.53$ | $1.9 \pm 0.52$ | $0.491 \pm 0.002$ |
| | 0.6 | $87.9 \pm 0.88$ | $\mathbf{17.6 \pm 1.64}$ | $\mathbf{0.516 \pm 0.008}$ | $91.1 \pm 2.4$ | $11.7 \pm 3.74$ | $0.54 \pm 0.018$ |
| | 0.7 | $85.5 \pm 0.95$ | $33.5 \pm 3.72$ | $0.514 \pm 0.021$ | $84.8 \pm 1.09$ | $\mathbf{36.1 \pm 0.98}$ | $\mathbf{0.502 \pm 0.007}$ |
| | 0.8 | $79.5 \pm 0.72$ | $57.1 \pm 2.1$ | $0.46 \pm 0.015$ | $76.3 \pm 1.33$ | $\mathbf{87.2 \pm 1.55}$ | $\mathbf{0.309 \pm 0.01}$ |
| | 0.83 | $75.8 \pm 1.61$ | $79.2 \pm 2.74$ | $0.368 \pm 0.016$ | $71.2 \pm 1.3$ | $\mathbf{99.8 \pm 0.13}$ | $\mathbf{0.289 \pm 0.013}$ |

Table 2: Performance results of NodeCwR-Cost with $LS = 0$ and $LS = \epsilon$. Results with all values of cost(d) are mentioned in Table 4

### 4.7 COVERAGE AND ACCURACY COMPARISONS BETWEEN NODECWR-COST AND NODECWR-COV

Here, we compare the performance of cost-based and coverage-based approaches. We do the comparison using coverage vs. accuracy plots. Figure 3 shows the coverage versus accuracy plots for both cost-based and coverage-based approaches on different datasets. The cost-based approach

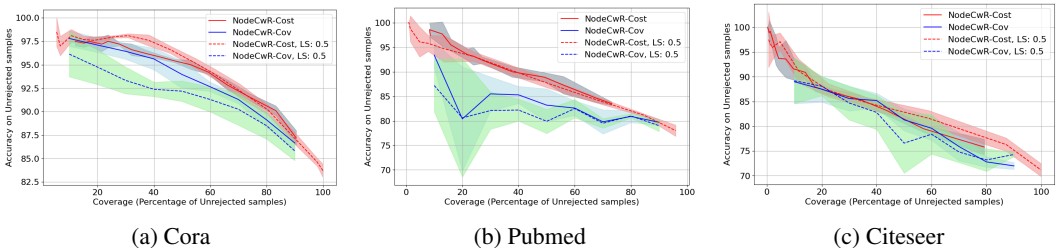

| (a) Cora | (b) Pubmed | (c) Citeseer |
| --- | --- | --- |

Figure 3: Coverage and Accuracy comparison between NodeCwR-Cost and NodeCwR-Cov

shows a clear advantage in terms of accuracy for most coverage rates. The reason is as follows. The coverage constraint in NodeCwR-Cov does not ensure the rejection of those examples that are hard to classify correctly. Thus, it may reject some of the easy examples. Thus, every coverage value may include more hard examples. Label smoothing makes this situation worse for NodeCwR-Cov due to soft labels. We also observe a very high standard deviation in the performance of NodeCwR-Cov. On the other hand, NodeCwR-Cost prefers to reject hard examples first by assigning a cost to rejection. This makes NodeCwR-Cost perform better than NodeCwR-Cov.

## 4.8 Node Embedding Visualisation

We plotted t-SNE plots to represent the predicted class of each node.

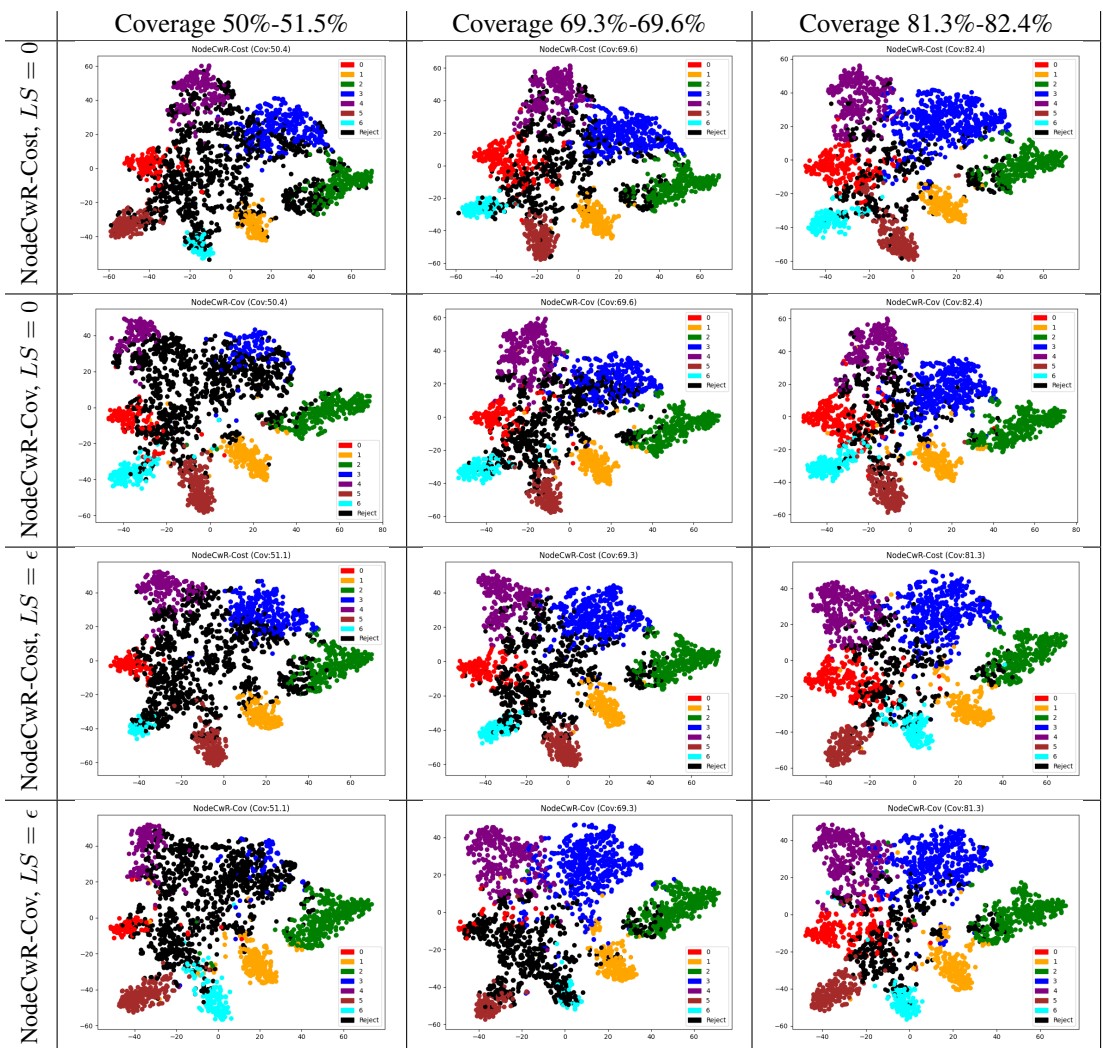

Figure 4: t-SNE plots representing predictions on Cora data set where black represents the reject option.

It was noticeable that in both models, the rejected examples are usually the nodes that highly overlap between two or more classes. We can also notice that as the model coverage decreases, the number of examples it rejects increases and covers more overlapping boundaries between classes. It is worth noting that although the coverage and accuracy are almost similar in both models, the examples that each model chooses to reject are from different overlapping classes.

## 4.9 Experiments With Label Noise

This section will discuss how label noise can affect the cost-based reject option classifiers learned for node classification. We consider symmetric label noise (10% and 20%). Figure 5 shows the effect of label noise on NodeCwR-Cost and NodeCwR-Cov in the absence of label smoothing. We vary the noise rate from 10% to 20%. We observe that the overall performance of both approaches worsens with the increase in the noise rate. Moreover, the coverage-based approach's performance degrades more than the cost-based approach with label noise.

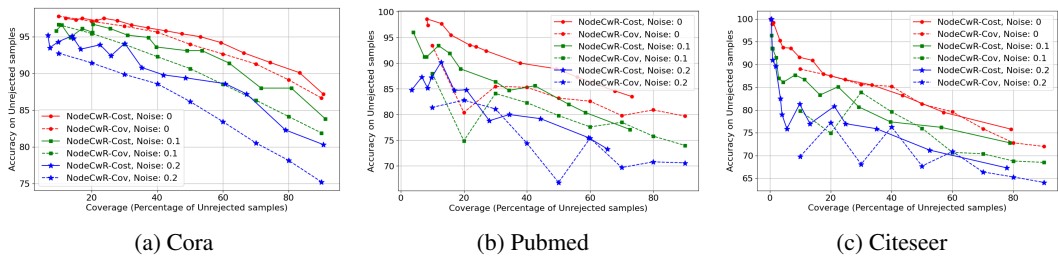

(a) Cora         (b) Pubmed         (c) Citeseer

Figure 5: Effect of label noise on NodeCwR-Cost and NodeCwR-Cov in the $LS = 0$ setting.

We performed experiments with label noise in the $LS = \epsilon$ setting to observe if label smoothing makes the learning robust. Figure 6 the results with label smoothing when there is label noise. We observe the following. In the cost-based models, when $LS = \epsilon$, increasing the noise rate decreases the performance. However, compared to $LS = 0$, $LS = \epsilon$ models show better performance. Thus, label smoothing brings robustness against label noise in cost-based models. In the coverage-based models, when $LS = \epsilon$, increasing the noise rate decreases the performance. However, label smoothing worsens the performance when there is label noise. Thus, label noise is ineffective in handling label noise in coverage-based models.

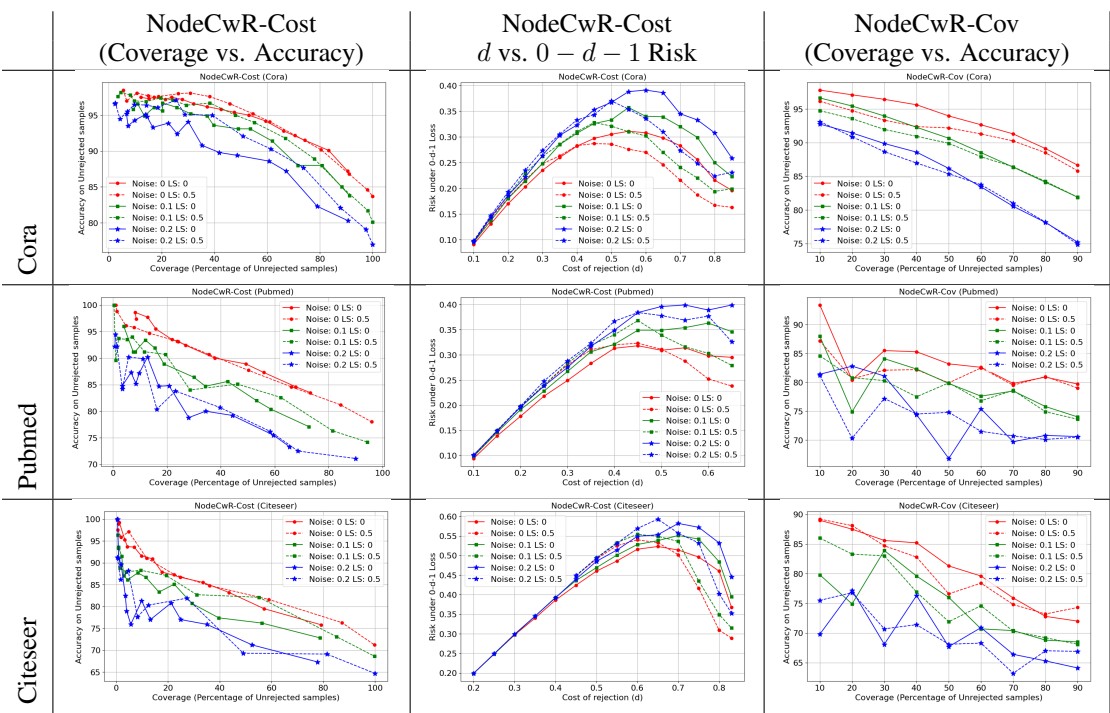

Figure 6: Effect of label noise on NodeCwR-Cost and NodeCwR-Cov in $LS = \epsilon$ setting. The actual numbers can be seen in Tables 3, 4, 5, and 6 in Appendix A and B.

## 5 CONCLUSION

We proposed a foundational model for node classification, which can abstain from making a prediction when uncertain. We set up abstention-based GAT architecture to learn node embeddings and show the effectiveness of two abstention models for graphs, which can be highly useful in high-risk applications. Our code implementation is added to the supplementary file.

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

# A   EXPERIMENTAL RESULTS OF NCWR-COV ON NOISY LABELS WITH LABEL SMOOTHING

| Dataset | Coverage | Noise=0.1, LS=0 | Noise=0.1, LS=0.5 | Noise=0.2, LS=0 | Noise=0.2, LS=0.5 |
|---------|----------|-----------------|-------------------|-----------------|-------------------|
| **Cora** | 0.1 | $96.63 \pm 2.387$ | $94.75 \pm 3.196$ | $92.75 \pm 2.55$ | $93.12 \pm 1.959$ |
| | 0.2 | $95.44 \pm 1.841$ | $93.56 \pm 2.597$ | $91.44 \pm 3.51$ | $90.88 \pm 3.091$ |
| | 0.3 | $93.96 \pm 2.229$ | $91.96 \pm 2.312$ | $89.87 \pm 3.06$ | $88.66 \pm 3.436$ |
| | 0.4 | $92.28 \pm 2.627$ | $90.94 \pm 1.898$ | $88.59 \pm 3.362$ | $86.97 \pm 3.687$ |
| | 0.5 | $90.65 \pm 2.463$ | $89.9 \pm 1.317$ | $86.17 \pm 3.152$ | $85.38 \pm 3.777$ |
| | 0.6 | $88.52 \pm 2.837$ | $87.9 \pm 1.095$ | $83.43 \pm 3.604$ | $83.75 \pm 3.411$ |
| | 0.7 | $86.36 \pm 2.344$ | $86.41 \pm 1.354$ | $80.54 \pm 3.232$ | $81.01 \pm 3.299$ |
| | 0.8 | $84.12 \pm 2.167$ | $84.25 \pm 1.501$ | $78.15 \pm 3.247$ | $78.2 \pm 2.909$ |
| | 0.9 | $81.89 \pm 2.071$ | $81.9 \pm 2.059$ | $75.22 \pm 2.906$ | $74.9 \pm 2.943$ |
| **Pubmed** | 0.1 | $88 \pm 8.92$ | $84.6 \pm 4.93$ | $81.4 \pm 7.67$ | $81.2 \pm 5.76$ |
| | 0.2 | $74.9 \pm 12.03$ | $80.8 \pm 15.09$ | $82.8 \pm 9.83$ | $70.3 \pm 8.93$ |
| | 0.3 | $84.1 \pm 3.88$ | $80.3 \pm 7.07$ | $81.1 \pm 2.9$ | $77.2 \pm 3.95$ |
| | 0.4 | $82.3 \pm 3.97$ | $77.5 \pm 4.55$ | $74.4 \pm 5.5$ | $74.5 \pm 2.89$ |
| | 0.5 | $79.8 \pm 1.82$ | $79.9 \pm 2.84$ | $66.8 \pm 4.22$ | $74.8 \pm 4.71$ |
| | 0.6 | $77.6 \pm 2.1$ | $76.8 \pm 3.62$ | $75.4 \pm 5.63$ | $71.5 \pm 5.55$ |
| | 0.7 | $78.5 \pm 1.88$ | $78.6 \pm 2.87$ | $69.7 \pm 3.43$ | $70.7 \pm 4.86$ |
| | 0.8 | $75.8 \pm 3.37$ | $74.9 \pm 2.06$ | $70.8 \pm 3.76$ | $70.1 \pm 3.9$ |
| | 0.9 | $74 \pm 1.83$ | $73.6 \pm 1.71$ | $70.6 \pm 1.39$ | $70.5 \pm 4.27$ |
| **Citeseer** | 0.1 | $79.8 \pm 8.14$ | $86 \pm 6.65$ | $69.8 \pm 24.56$ | $75.5 \pm 7.11$ |
| | 0.2 | $74.9 \pm 14.26$ | $83.3 \pm 2.97$ | $77.2 \pm 4.47$ | $76.8 \pm 5.83$ |
| | 0.3 | $83.9 \pm 3.86$ | $83 \pm 4.15$ | $68.1 \pm 5.17$ | $70.7 \pm 6.68$ |
| | 0.4 | $79.6 \pm 2.36$ | $76.9 \pm 5.33$ | $76.3 \pm 2.63$ | $71.4 \pm 4.71$ |
| | 0.5 | $76 \pm 2.35$ | $71.9 \pm 7.83$ | $67.7 \pm 5.51$ | $68.1 \pm 4.39$ |
| | 0.6 | $70.7 \pm 2.4$ | $74.6 \pm 2.01$ | $70.9 \pm 3.77$ | $68.3 \pm 3.78$ |
| | 0.7 | $70.4 \pm 2.86$ | $70.3 \pm 4.42$ | $66.4 \pm 4.13$ | $63.2 \pm 8.25$ |
| | 0.8 | $68.8 \pm 3.07$ | $69.2 \pm 4.08$ | $65.3 \pm 3.12$ | $67 \pm 2.3$ |
| | 0.9 | $68.5 \pm 3.47$ | $68.1 \pm 2.6$ | $64.1 \pm 3.09$ | $66.9 \pm 4.89$ |

Table 3: Accuracy and Coverage of NCwR-Cov on noisy labels with Label Smoothing.

# B  EXPERIMENTAL RESULTS OF NCwR-COST WITH LABEL SMOOTHING

| | $d$ | Noise rate=0 | | | | | |
|---|---|---|---|---|---|---|---|
| | | LS=0 | | | LS=0.5 | | |
| | | Acc. on Unrejected | Coverage | 0-d-1 Risk | Acc. on Unrejected | Coverage | 0-d-1 Risk |
| **Cora** | 0.1 | 97.5 ± 0.48 | **12.1 ± 0.95** | **0.091 ± 0.001** | 98.5 ± 1.52 | 5.5 ± 0.97 | 0.095 ± 0.001 |
| | 0.15 | 97.3 ± 0.42 | **15.2 ± 1.29** | **0.131 ± 0.002** | 97 ± 1.03 | 6.8 ± 0.74 | 0.142 ± 0.001 |
| | 0.2 | 97.5 ± 0.47 | **17.1 ± 1.6** | **0.17 ± 0.003** | 98.1 ± 0.68 | 10.7 ± 1.14 | 0.181 ± 0.003 |
| | 0.25 | 97.2 ± 0.79 | **21.5 ± 2.31** | **0.203 ± 0.004** | 97.7 ± 0.38 | 15 ± 0.68 | 0.216 ± 0.002 |
| | 0.3 | 97.5 ± 0.68 | **23.8 ± 1.75** | **0.235 ± 0.005** | 97.6 ± 0.38 | 18.8 ± 2.41 | 0.248 ± 0.007 |
| | 0.35 | 97.2 ± 0.18 | **27.8 ± 1.73** | **0.26 ± 0.005** | 98 ± 0.12 | 26.3 ± 2.42 | 0.263 ± 0.008 |
| | 0.4 | 96.6 ± 0.62 | **32.2 ± 0.91** | **0.282 ± 0.004** | 98.1 ± 0.21 | 30.9 ± 2.09 | 0.283 ± 0.008 |
| | 0.45 | 96.2 ± 0.37 | 37.2 ± 2.36 | 0.297 ± 0.01 | 97.6 ± 0.59 | **38.2 ± 2.01** | **0.287 ± 0.007** |
| | 0.5 | 95.8 ± 0.23 | 42.6 ± 1.88 | 0.305 ± 0.009 | 96.6 ± 0.39 | **45.9 ± 1.53** | **0.286 ± 0.007** |
| | 0.55 | 95.4 ± 0.18 | 47.5 ± 1.12 | 0.311 ± 0.005 | 95.2 ± 0.51 | **54.6 ± 1.74** | **0.276 ± 0.007** |
| | 0.6 | 95 ± 0.49 | 53.1 ± 1.07 | 0.308 ± 0.006 | 94.1 ± 0.55 | **61 ± 0.4** | **0.27 ± 0.003** |
| | 0.65 | 94.2 ± 0.21 | 59.4 ± 1.87 | 0.298 ± 0.01 | 92.2 ± 0.73 | **70.6 ± 1.49** | **0.246 ± 0.005** |
| | 0.7 | 92.8 ± 0.55 | 66.3 ± 1.81 | 0.283 ± 0.009 | 90.2 ± 0.52 | **80.4 ± 0.82** | **0.216 ± 0.006** |
| | 0.75 | 91.5 ± 0.39 | 74.3 ± 0.87 | 0.256 ± 0.005 | 86.8 ± 0.61 | **91.1 ± 1.05** | **0.187 ± 0.008** |
| | 0.8 | 90.1 ± 0.51 | 83.3 ± 0.68 | 0.216 ± 0.008 | 84.6 ± 0.42 | **98 ± 0.34** | **0.167 ± 0.003** |
| | 0.85 | 87.2 ± 0.78 | 90.5 ± 1.11 | 0.196 ± 0.003 | 83.7 ± 0.7 | **100 ± 0.04** | **0.163 ± 0.007** |
| | 1 | 82.4 ± 0.42 | **100** | **0.176 ± 0.004** | 82.2 ± 0.4 | **100** | 0.178 ± 0.004 |
| **Pubmed** | 0.1 | 97.4 ± 1.86 | **8.6 ± 3.12** | **0.094 ± 0.001** | 100 | 0.6 ± 0.25 | 0.099 |
| | 0.15 | 98.6 ± 1.4 | **8.3 ± 2.79** | **0.139 ± 0.003** | 100 | 1 ± 0.79 | 0.148 ± 0.001 |
| | 0.2 | 97.7 ± 2.49 | **12.9 ± 4.52** | **0.178 ± 0.005** | 98.8 ± 2.63 | 1.5 ± 0.47 | 0.197 ± 0.001 |
| | 0.25 | 95.5 ± 1.21 | **15.9 ± 2.07** | **0.218 ± 0.003** | 96.1 ± 3.05 | 4.8 ± 0.9 | 0.24 ± 0.003 |
| | 0.3 | 93.5 ± 1.2 | **21.9 ± 2.77** | **0.249 ± 0.007** | 95.8 ± 1.62 | 7.9 ± 3.08 | 0.28 ± 0.008 |
| | 0.35 | 93.2 ± 1.8 | **23.9 ± 4.89** | **0.283 ± 0.011** | 94.7 ± 2.69 | 13.5 ± 3.03 | 0.31 ± 0.008 |
| | 0.4 | 92.4 ± 0.66 | **27 ± 5.33** | **0.313 ± 0.016** | 93.1 ± 0.95 | 24.2 ± 1.22 | 0.32 ± 0.005 |
| | 0.45 | 90 ± 1.08 | **37.8 ± 2.91** | **0.318 ± 0.007** | 90.7 ± 1.03 | 35.6 ± 1.91 | 0.323 ± 0.008 |
| | 0.5 | 88.9 ± 0.92 | 49.3 ± 5.48 | **0.309 ± 0.018** | 87.7 ± 0.94 | **50.2 ± 3.36** | 0.311 ± 0.01 |
| | 0.55 | 87.3 ± 1.77 | 55.9 ± 6.91 | 0.314 ± 0.02 | 84.6 ± 0.67 | **66.1 ± 4.13** | **0.288 ± 0.017** |
| | 0.6 | 84.6 ± 0.87 | 67.8 ± 4.81 | 0.298 ± 0.019 | 81.2 ± 0.29 | **84.4 ± 3.5** | **0.252 ± 0.015** |
| | 0.65 | 83.5 ± 0.36 | 73.2 ± 1.67 | 0.295 ± 0.008 | 78 ± 1.14 | **95.9 ± 0.47** | **0.238 ± 0.01** |
| | 1 | 77.1 ± 0.29 | **100** | 0.229 ± 0.003 | 77.2 ± 0.94 | **100** | **0.228 ± 0.009** |
| **Citeseer** | 0.1 | - | **0** | **0.1** | - | **0** | **0.1** |
| | 0.15 | 100 | **0.6 ± 0.42** | **0.149 ± 0.001** | - | 0 | 0.15 |
| | 0.2 | 100 | **0.3 ± 0.17** | **0.199** | - | 0 | 0.2 |
| | 0.25 | 98.9 ± 2.49 | **0.8 ± 0.67** | **0.248 ± 0.001** | - | 0 | 0.25 |
| | 0.3 | 99.2 ± 1.72 | **1.2 ± 0.85** | **0.297 ± 0.002** | - | 0 | 0.3 |
| | 0.35 | 95.3 ± 2.91 | **3.3 ± 0.79** | **0.34 ± 0.002** | - | 0 | 0.35 |
| | 0.4 | 93.7 ± 2.26 | **4.3 ± 1.18** | **0.386 ± 0.004** | - | 0 | 0.4 |
| | 0.45 | 93.6 ± 3.76 | **6.9 ± 1.74** | **0.424 ± 0.005** | 97.5 ± 5.59 | 0.6 ± 0.25 | 0.448 ± 0.001 |
| | 0.5 | 91.6 ± 2.23 | **9.7 ± 1.7** | **0.46 ± 0.005** | 95.9 ± 2.53 | 1.9 ± 0.52 | 0.491 ± 0.002 |
| | 0.55 | 90.9 ± 1.33 | **14 ± 1.28** | **0.486 ± 0.007** | 97.1 ± 1.77 | 4.8 ± 0.49 | 0.525 ± 0.003 |
| | 0.6 | 87.9 ± 0.88 | **17.6 ± 1.64** | **0.516 ± 0.008** | 91.1 ± 2.4 | 11.7 ± 3.74 | 0.54 ± 0.018 |
| | 0.65 | 86.7 ± 0.7 | **24.7 ± 1.58** | **0.523 ± 0.007** | 87.3 ± 1.32 | 22.4 ± 1.68 | 0.533 ± 0.01 |
| | 0.7 | 85.5 ± 0.95 | 33.5 ± 3.72 | 0.514 ± 0.021 | 84.8 ± 1.09 | **36.1 ± 0.98** | **0.502 ± 0.007** |
| | 0.75 | 83.2 ± 0.96 | 43.6 ± 2.13 | 0.496 ± 0.011 | 81.6 ± 1.58 | **59 ± 2.21** | **0.416 ± 0.02** |
| | 0.8 | 79.5 ± 0.72 | 57.1 ± 2.1 | 0.46 ± 0.015 | 76.3 ± 1.33 | **87.2 ± 1.55** | **0.309 ± 0.01** |
| | 0.83 | 75.8 ± 1.61 | 79.2 ± 2.74 | 0.368 ± 0.016 | 71.2 ± 1.3 | **99.8 ± 0.13** | **0.289 ± 0.013** |
| | 1 | 70.3 ± 0.5 | **100** | 0.297 ± 0.005 | 70.8 ± 0.44 | **100** | **0.292 ± 0.004** |

Table 4: Performance results of NodeCwR-Cost with $LS = 0$ and $LS = \epsilon$.

| | | Noise rate=0.1 | | | | | |
|---|---|---|---|---|---|---|---|
| | | LS=0 | | | LS=0.5 | | |
| | $d$ | Acc. on Unrejected | Coverage | 0-d-1 Risk | Acc. on Unrejected | Coverage | 0-d-1 Risk |
| **Cora** | 0.1 | 95.8 ± 1.27 | 9.2 ± 1.6 | 0.095 ± 0.001 | 97.6 ± 2.16 | 3.4 ± 1.02 | 0.097 ± 0.001 |
| | 0.15 | 96.6 ± 1.45 | 10.9 ± 0.72 | 0.137 ± 0.002 | 98.2 ± 1.07 | 4.6 ± 1.04 | 0.144 ± 0.001 |
| | 0.2 | 94.9 ± 1.96 | 13.5 ± 1.85 | 0.18 ± 0.003 | 97.8 ± 0.9 | 8.1 ± 1.04 | 0.186 ± 0.002 |
| | 0.25 | 96.1 ± 1.09 | 17.1 ± 1.32 | 0.214 ± 0.002 | 97 ± 1.46 | 9.7 ± 2.22 | 0.229 ± 0.005 |
| | 0.3 | 95.6 ± 0.68 | 20.3 ± 1.63 | 0.248 ± 0.005 | 96.9 ± 0.53 | 14.1 ± 1.29 | 0.262 ± 0.003 |
| | 0.35 | 96.7 ± 1.33 | 20.4 ± 1.91 | 0.285 ± 0.007 | 97.4 ± 0.66 | 19.6 ± 2.19 | 0.286 ± 0.007 |
| | 0.4 | 96.1 ± 0.88 | 25.7 ± 2.11 | 0.307 ± 0.008 | 97.1 ± 0.62 | 24.2 ± 4 | 0.31 ± 0.014 |
| | 0.45 | 95.2 ± 0.82 | 31 ± 1.78 | 0.326 ± 0.005 | 96.4 ± 0.95 | 29.4 ± 2.55 | 0.328 ± 0.01 |
| | 0.5 | 94.9 ± 1.2 | 37.3 ± 4.04 | 0.333 ± 0.02 | 96.7 ± 0.53 | 38.3 ± 2.69 | 0.321 ± 0.014 |
| | 0.55 | 93.6 ± 0.84 | 39.7 ± 3.91 | 0.357 ± 0.021 | 95 ± 0.85 | 48 ± 4.89 | 0.31 ± 0.025 |
| | 0.6 | 93.1 ± 1.91 | 49 ± 3.61 | 0.34 ± 0.022 | 94 ± 0.63 | 55.1 ± 2.34 | 0.302 ± 0.012 |
| | 0.65 | 93.1 ± 1.78 | 53.6 ± 1.72 | 0.339 ± 0.01 | 91.8 ± 1.2 | 66.9 ± 2.01 | 0.27 ± 0.015 |
| | 0.7 | 91.4 ± 1.11 | 61.9 ± 4.48 | 0.32 ± 0.023 | 88.9 ± 1.11 | 77.9 ± 1.99 | 0.241 ± 0.014 |
| | 0.75 | 88 ± 1.04 | 71.6 ± 3.93 | 0.299 ± 0.03 | 85 ± 1.4 | 88.3 ± 1.47 | 0.22 ± 0.016 |
| | 0.8 | 88 ± 1.46 | 80.8 ±1.62 | 0.25 ± 0.015 | 81.7 ± 2.35 | 98.2 ± 0.82 | 0.194 ± 0.02 |
| | 0.85 | 83.8 ± 1.78 | 91.2 ±0.83 | 0.223 0± 0.011 | 80.1 ± 2.21 | 100 | 0.199 ± 0.022 |
| | 1 | 77.3 ± 0.98 | 100 | 0.227 ± 0.01 | 77.5 ± 5.74 | 100 | 0.225 ± 0.058 |
| **Pubmed** | 0.1 | 96 ± 3.33 | 4 ± 1.74 | 0.098 ± 0.001 | 100 | 0.4 ± 0.26 | 0.1 |
| | 0.15 | 91.2 ± 3.14 | 7.5 ± 2.37 | 0.146 ± 0.002 | 89.6 ± 12.2 | 1 ± 0.6 | 0.15 ± 0.001 |
| | 0.2 | 91.2 ± 0.77 | 8.2 ± 1.65 | 0.191 ± 0.002 | 93.7 ± 7.95 | 2.1 ± 1.05 | 0.198 ± 0.002 |
| | 0.25 | 93.4 ± 2.82 | 12 ± 4.25 | 0.228 ± 0.008 | 93.5 ± 2.38 | 5.3 ± 1.32 | 0.24 ± 0.003 |
| | 0.3 | 91.9 ± 2.11 | 15.5 ± 3.95 | 0.267 ± 0.007 | 94 ± 2.57 | 7.1 ± 1.99 | 0.283 ± 0.003 |
| | 0.35 | 88.9 ± 2.75 | 18.9 ± 9.44 | 0.306 ± 0.018 | 91.2 ± 2.46 | 11.7 ± 1.97 | 0.319 ± 0.007 |
| | 0.4 | 86.4 ± 2.28 | 30 ± 4.1 | 0.321 ± 0.011 | 90.7 ± 1.7 | 19.5 ± 5.34 | 0.34 ± 0.017 |
| | 0.45 | 84.7 ± 2.78 | 34.3 ± 2.33 | 0.349 ± 0.009 | 84 ± 4.36 | 28.6 ± 2.89 | 0.368 ± 0.01 |
| | 0.5 | 85.6 ± 2.43 | 42.5 ± 4.29 | 0.349 ± 0.017 | 85.1 ± 2.21 | 46.2 ± 4.52 | 0.339 ± 0.009 |
| | 0.55 | 82 ± 2.29 | 53.2 ± 5.56 | 0.354 ± 0.017 | 82.6 ± 2.3 | 62.2 ± 5.59 | 0.316 ± 0.03 |
| | 0.6 | 80.4 ± 1.48 | 58.5 ± 4.24 | 0.363 ± 0.025 | 76.3 ± 2.32 | 81.5 ± 4.47 | 0.303 ± 0.032 |
| | 0.65 | 77.1 ± 3.33 | 72.6 ± 7.61 | 0.346 ± 0.023 | 74.2 ± 3.67 | 94.3 ± 2.38 | 0.279 ± 0.04 |
| | 1 | 70.3 ± 1.18 | 100 | 0.297 ± 0.012 | 73.2 ± 3.19 | 100 | 0.268 ± 0.032 |
| **Citeseer** | 0.1 | - | - | - | - | - | - |
| | 0.15 | - | - | - | - | - | - |
| | 0.2 | 96.4± 8.13 | 0.6 ± 0.37 | 0.199 ±0.001 | - | - | - |
| | 0.25 | 93.6 ±6.3 | 0.8 ± 0.34 | 0.248 ±0.001 | - | - | - |
| | 0.3 | 93.5 ±6.15 | 0.9 ± 0.54 | 0.298± 0.001 | - | - | - |
| | 0.35 | 91.5± 5.61 | 2.1 ± 1.03 | 0.345 ±0.003 | - | - | - |
| | 0.4 | 87 ± 6.59 | 2.9 ± 1 | 0.393± 0.001 | - | - | - |
| | 0.45 | 86.1± 7.48 | 4.4 ± 0.43 | 0.436± 0.004 | 93.3± 14.9 | 0.8 ± 0.73 | 0.447 ±0.004 |
| | 0.5 | 87.7± 2.65 | 8.3 ± 2.26 | 0.469± 0.007 | 88.8± | 5.2 ± 1.4 0.66 | 0.494± 0.003 |
| | 0.55 | 86.7 ±6.75 | 11.4 ±3.82 | 0.501 ±0.02 | 87.9 ±3.13 | 3.8 ± 1.26 | 0.533 ±0.006 |
| | 0.6 | 83.3 ±5.14 | 16.5 ±0.95 | 0.528± 0.011 | 88.3 ±1.56 | 9.4 ± 1.18 | 0.554 ±0.006 |
| | 0.65 | 85.1± 1.91 | 22.4 ±2.87 | 0.538 ±0.017 | 87.1 ±0.93 | 19.3 ±2.72 | 0.549± 0.014 |
| | 0.7 | 80.7± 1.96 | 29.2 ±2.54 | 0.551± 0.017 | 82.7± 3.4 | 31.1± 4.08 | 0.536± 0.021 |
| | 0.75 | 77.4± 1.94 | 39.6± 2.14 | 0.542 ±0.014 | 82.1 ±2.28 | 55.1 ±4.71 | 0.435± 0.034 |
| | 0.8 | 76.2± 2.11 | 56.3 ±3.56 | 0.484± 0.027 | 73.1± 2.55 | 85.1± 2.74 | 0.349± 0.016 |
| | 0.83 | 72.8 ±1.27 | 78.8 ±5.09 | 0.395 ±0.038 | 68.6± 2.39 | 99.8± 0.25 | 0.315± 0.024 |
| | 1 | 66.8 ±2.35 | 100 ± | 0.332± 0.0235 | 67.4 ±2.25 | 100± | 0.326 ±0.023 |

Table 5: Performance results of NodeCwR-Cost with $LS = 0$ and $LS = \epsilon$ on noise 0.1

| | | Noise rate=0.2 | | | | | |
|---|---|---|---|---|---|---|---|
| | | **LS=0** | | | **LS=0.5** | | |
| | $d$ | **Acc. on Unrejected** | **Coverage** | **0-d-1 Risk** | **Acc. on Unrejected** | **Coverage** | **0-d-1 Risk** |
| Cora | 0.1 | 95.2 ±1.68 | 6.7 ±2.05 | 0.096 ±0.002 | 96.6 ±4.08 | 2.3 ±0.87 | 0.098 ±0.001 |
| | 0.15 | 93.5 ±2.19 | 7.3 ±0.69 | 0.144 ±0.001 | 96.7 ±3.15 | 2.5 ±0.58 | 0.147 ±0.001 |
| | 0.2 | 94.3 ±1.64 | 9.8 ±1.42 | 0.186 ±0.002 | 94.5 ±4.61 | 4.2 ±1.6 | 0.193 ±0.004 |
| | 0.25 | 95.1 ±3.1 | 14 ±1.53 | 0.222 ±0.005 | 95.5 ±3.67 | 7.1 ±1.26 | 0.235 ±0.004 |
| | 0.3 | 94.8 ±0.95 | 14.5 ±2.76 | 0.264 ±0.007 | 96.5 ±1.68 | 9.7 ±2.02 | 0.274 ±0.007 |
| | 0.35 | 93.3 ±1.71 | 16.7 ±1.21 | 0.303 ±0.004 | 96.4 ±1.09 | 14.2 ±3.35 | 0.305 ±0.012 |
| | 0.4 | 93.9 ±1.24 | 22.6 ±2.3 | 0.323 ±0.010 | 96.1 ±1 | 18.6 ±1.9 | 0.333 ±0.007 |
| | 0.45 | 92.4 ±1.66 | 25.9 ±2.69 | 0.353 ±0.010 | 97.1 ±0.89 | 25.5 ±2.8 | 0.343 ±0.01 |
| | 0.5 | 94.1 ±0.56 | 30.1 ±3.19 | 0.367 ±0.015 | 95.1 ±1.13 | 28.8 ±5.19 | 0.37 ±0.03 |
| | 0.55 | 90.8 ±2.26 | 35.3 ±4.13 | 0.388 ±0.023 | 95 ±1.29 | 39.2 ±3.37 | 0.354 ±0.015 |
| | 0.6 | 89.8 ±1.74 | 41.9 ±3.4 | 0.391 ±0.017 | 92.1 ±0.93 | 50.8 ±3.72 | 0.336 ±0.012 |
| | 0.65 | 89.4 ±2.5 | 48.6 ±4.73 | 0.386 ±0.029 | 90.3 ±1.93 | 61.4 ±2.34 | 0.31 ±0.02 |
| | 0.7 | 88.6 ±2.48 | 60.7 ±4.5 | 0.345 ±0.012 | 87.7 ±1.16 | 73.8 ±3.51 | 0.274 ±0.019 |
| | 0.75 | 87.2 ±2.45 | 67.2 ±4.52 | 0.333 ±0.017 | 82.1 ±1.97 | 87.8 ±3.31 | 0.249 ±0.012 |
| | 0.8 | 82.3 ±2.59 | 79 ±1.6 | 0.308 ±0.027 | 79.1 ±1.59 | 97.4 ±0.33 | 0.224 ±0.015 |
| | 0.85 | 80.3 ±2.94 | 90.6 ±2.04 | 0.259 ±0.023 | 77 ±3.17 | 100 ±0.04 | 0.231 ±0.032 |
| | 1 | 71.9 ±2.36 | 100 | 0.281 ±0.023 | 73.6 ±3.1 | 100 | 0.264 ±0.031 |
| Pubmed | 0.1 | 84.8 ±8.27 | 3.5 ±1.07 | 0.101 ±0.002 | 92.2 ±14.4 | 0.7 ±0.64 | 0.1 ±0.001 |
| | 0.15 | 87.3 ±3.05 | 6.7 ±3.39 | 0.149 ±0.002 | 94.5 ±12.2 | 0.9 ±0.76 | 0.149 ±0.002 |
| | 0.2 | 85.2 ±4.61 | 8.5 ±3.41 | 0.196 ±0.004 | 92.2 ±7.54 | 1.5 ±0.65 | 0.198 ±0.001 |
| | 0.25 | 87.2 ±2.19 | 9.8 ±3.32 | 0.238 ±0.006 | 84.2 ±11.8 | 3.5 ±3.51 | 0.248 ±0.003 |
| | 0.3 | 90.2 ±5.48 | 12.8 ±2.83 | 0.275 ±0.005 | 90.2 ±4.54 | 5.8 ±0.94 | 0.288 ±0.004 |
| | 0.35 | 84.7 ±2.68 | 17 ±3.63 | 0.317 ±0.008 | 89.9 ±3.37 | 11.1 ±2.87 | 0.323 ±0.007 |
| | 0.4 | 84.8 ±4.75 | 20.8 ±5.36 | 0.349 ±0.013 | 80.4 ±8.24 | 16.3 ±3.08 | 0.367 ±0.015 |
| | 0.45 | 78.8 ±6.32 | 28.1 ±5.13 | 0.384 ±0.02 | 83.8 ±4.15 | 23.2 ±5.02 | 0.384 ±0.016 |
| | 0.5 | 80 ±3.25 | 34.4 ±8.37 | 0.396 ±0.032 | 80.7 ±1.49 | 39.7 ±3.03 | 0.378 ±0.009 |
| | 0.55 | 79.2 ±4.55 | 44.3 ±6.56 | 0.399 ±0.029 | 76.2 ±7.03 | 58.3 ±4.75 | 0.369 ±0.039 |
| | 0.6 | 75.5 ±3.39 | 59.5 ±5.98 | 0.389 ±0.029 | 72.5 ±2.77 | 68.6 ±10.42 | 0.377 ±0.04 |
| | 0.65 | 73.3 ±6.09 | 65.5 ±2.91 | 0.399 ±0.042 | 71.1 ±3.93 | 90 ±5 | 0.326 ±0.034 |
| | 1 | 64.1 ±4.95 | 100 ±0 | 0.359 ±0.05 | 65.5 ±3.74 | 100 0 | 0.345 ±0.037 |
| Citeseer | 0.1 | - | - | - | - | - | - |
| | 0.15 | - | - | - | - | - | - |
| | 0.2 | 100 | 0.4 ±0.2 | 0.199 ±0.001 | - | - | - |
| | 0.25 | 100 | 0.5 ±0.15 | 0.249 ±0.001 | - | - | - |
| | 0.3 | 91 ±14.51 | 0.9 ±0.6 | 0.299 ±0.001 | - | - | - |
| | 0.35 | 89.6 ±8.95 | 1.9 ±0.95 | 0.345 ±0.003 | - | - | - |
| | 0.4 | 82.5 ±5.38 | 3.5 ±1.21 | 0.392 ±0.004 | - | - | - |
| | 0.45 | 79 ±10.23 | 4 ±1.01 | 0.44 ±0.006 | 91.2 ±11.8 | 0.4 ±0.4 | 0.449 ±0.001 |
| | 0.5 | 75.9± 2.53 | 5.6 ±1 | 0.485 ±0.003 | 86.2 ±15.2 | 1.7 ±1.6 | 0.493 ±0.007 |
| | 0.55 | 81.3 ±2.11 | 9.8 ±2.63 | 0.514 ±0.01 | 88.2 ±6.01 | 4.6 ±2.5 | 0.531 ±0.008 |
| | 0.6 | 77 ±7.54 | 13.2 ±4.36 | 0.549 ±0.026 | 77.6 ±6.38 | 8.2 ±0.86 | 0.569 ±0.007 |
| | 0.65 | 80.8 ±2.35 | 21.2 ±3.53 | 0.553 ±0.019 | 80.3 ±2.37 | 12.4 ±4.05 | 0.593 ±0.019 |
| | 0.7 | 77 ±4.45 | 24.9 ±2.95 | 0.582 ±0.022 | 81.9 ±2.72 | 27.3 ±5.84 | 0.557 ±0.038 |
| | 0.75 | 75.9 ±2.03 | 35 ±6.5 | 0.572 ±0.033 | 69.3 ±9 | 49.1 ±5.48 | 0.532 ±0.05 |
| | 0.8 | 71.2 ±3.87 | 52.5 ±2.85 | 0.532 ±0.014 | 69.1 ±5.66 | 81.4 ±3.46 | 0.402 ±0.034 |
| | 0.83 | 67.3 ±6.49 | 77.8 ±3.85 | 0.445 ±0.035 | 64.7 ±3.79 | 99.9 ±0.19 | 0.353 ±0.038 |
| | 1 | 59 ±4.09 | 100 | 0.41 ±0.041 | 63.7 ±5.07 | 100 | 0.373 ±0.0507 |

Table 6: Performance results of NodeCwR-Cost with $LS = 0$ and $LS = \epsilon$ and noise 0.2

