# OpenReview forum: "Node-CwR: Node Classification with Reject Option"
_ICLR.cc/2024/Conference — ICLR 2024 Conference Withdrawn Submission_

### Official Review · Reviewer_JNJ6 · 2023-10-30

**Soundness:** 2 fair
**Presentation:** 3 good
**Contribution:** 2 fair
**Rating:** 3
**Confidence:** 4

**Summary:**

The paper addresses the node classification task and extends it to consider classification-with-reject. The paper provides both cost-based and coverage-based models. Experiments on three small datasets provide insights into the behaviour of the proposed approaches. The experiments also investigate the impact of label noise and show that label smoothing is effective for the derived cost-based model.

**Strengths:**

S1.	The node classification with reject task has received little if any prior attention in the literature.
S2.	The experiments show that the proposed techniques offer promising performance and provide insights into their behaviour.

**Weaknesses:**

W1.	The technical contribution seems limited. The approaches are very close to existing CwR methods (developed for the non-graph setting); it’s difficult to see how the graph has posed an additional, meaningful challenge.
W2.	The experiments are conducted for three small graphs; papers on graph learning really need to go beyond Cora, Citeseer, and Pubmed – there are many benchmark datasets available now. The expectation is that experiments would be conducted with 6-8 datasets, with several being medium- to large- scale. One might also expect experiments in both supervised and semi-supervised settings, and both transductive and inductive.
W3.	The experiments do not compare to any baseline methods. While there may not be prior work that directly addresses this problem, I think it is relatively easy to construct a naïve baseline. A simple baseline would be training a standard node-classifier (ignoring the regret option) to derive embeddings and then using those embeddings in the standard non-graph CwR framework to train an MLP architecture. Another basic baselines would involve rejecting nodes according to a threshold on softmax entropy.

W1 (cont.): The main weakness of the paper is that there is a limited technical contribution. It’s hard to see how the coverage-based classifier differs from SelectiveNet beyond introducing a GAT, which is not a substantial technical innovation. The cost-based approach follows Cao et al. 2022 closely; the only extension seems to be the introduction of label smoothing. The paper needs to make it much clearer what technical challenge arises because of the presence of a graph and how that has led to design differences and innovations. The replacement of a non-graph classifier with a GAT is not enough.

**Questions:**

Q1.	Please provide a clearer explanation of how the presence of the graph has a significant impact on the CwR methodology and identify the main technical contributions and innovations of the paper. Please explain why they are important, novel, and substantial.
Q2.	Why is it sufficient to conduct experiments on only three small graphs? How do we know that the observations extend to graphs from different domains? How do we know that the same observations apply for larger scale graphs? Do the results also apply to supervised settings? What about the inductive setting?
Q3.	Why is it not possible to construct a naïve baseline for comparison, using any graph-learning technique to derive embeddings and then treating the problem using the standard CwR approach?

---

### Official Review · Reviewer_C48L · 2023-10-31

**Soundness:** 2 fair
**Presentation:** 2 fair
**Contribution:** 2 fair
**Rating:** 3
**Confidence:** 4

**Summary:**

The paper proposed a new approach called Node-CwR, which models node classification with a reject option using GAT. Two different models are proposed, cost-based and coverage-based. Empirically, the paper shows the effectiveness of the proposed models in learning efficient reject option models for node classification tasks.

**Strengths:**

1. The idea of investigating integrating reject option in node classification task is interesting.
2. Reproducible as the source code is attached.
3. Writing is clear and easy to follow.

**Weaknesses:**

1. The first sentence of abstract – “Graph attention networks (GAT) have been state-of-the-art GNN architecture used as the backbone for various graph learning problems” is not convinced. As far as I know, most of SOTA GNNs in node classification are not based on GAT.
2. The novelty is limited: just simply combine reject option and GAT. It is unclear why only use GAT as backbone. And it is unclear what is the specific design for graph data.
3. Although there are some related works of reject option classification in Section 2.1, there is no comparison between the proposed method and existing method.
4. The experiment is conducted on only three small datasets, which is not enough.
5. The notation is not well clarified. For example, the first equation in Section 3.1, I can't find any explanation to what is $S_n$.
6. The typesetting needs improvement for better readability. Lots of tables and figures are overfull.
7.  Figures 1 and 2 are notably blurry and similar to each other. Consequently, it is advisable to consolidate these two figures and make it clearer.

**Questions:**

1. Why only use GAT as the architecture? Can the proposed method benefit other GNN architecture?
2. What is the difference of the proposed reject option node classification compared to existing reject option classification?
3. Why experiment is only conducted on three datasets? How effective is the proposed method when applied to larger graphs or heterophilic graphs?

---

### Official Review · Reviewer_Kn9X · 2023-11-05

**Soundness:** 3 good
**Presentation:** 3 good
**Contribution:** 2 fair
**Rating:** 3
**Confidence:** 4

**Summary:**

The authors proposed two methods for node classification with reject option that can be applied to the graph attention networks. The coverage-based model takes the coverage as input and finds the optimal model for a given coverage rate. The cost-based model finds the optimal classifier for a given cost of rejection value. The authors then demonstrate the performance of the methods in multiple datasets under several hyperparameter settings, including label smoothing parameters.

**Strengths:**

Strengths:
- Interesting application of classification with reject option on GAT architecture.
- The paper is easy to understand.
- The author provides a detailed description of the experiment results.
- The authors also study the effect of label smoothing on the experiments.

**Weaknesses:**

Weakness:
1) Both methods presented in the paper are heavily influenced by previous research. The coverage model is based on SelectiveNet (Geifman & El-Yaniv, 2019), whereas the cost-based model is based on (Cao et al., 2022). The authors applied the previous research to the GAT learning setting.
2) The authors did not provide baselines for comparison in the experiments sections.
3) In summary, I think the authors provide a nice study on the application of classification with reject option to node classification with graph attention networks. However, I think ICLR may not be the best venue for this work.

**Questions:**

Questions:
1) The authors mentioned that the approaches work on GAT. However, I don't see any limitation that restricts the application of the proposed approach to other architectures. How do the approaches extend to other architectures?
2) The authors specifically mentioned the number of nodes in the architecture design. Why this specific number?

---

### Official Review · Reviewer_A6Y2 · 2023-11-09

**Soundness:** 2 fair
**Presentation:** 2 fair
**Contribution:** 2 fair
**Rating:** 3
**Confidence:** 4

**Summary:**

This paper addresses the problem of node classification with a reject option. The authors base their framework on the foundational Graph Attention Network (GAT), a prevalent graph neural network for graph embedding. To enable the reject option, they introduce a model called Node-CwR, which comprises two key modules: a cost-based model and a coverage-based model. Through a series of experiments conducted on various benchmark datasets, the authors showcase the effectiveness of their proposed model.

**Strengths:**

1. The paper explores an intriguing perspective – the reject option – which represents an interesting approach to node classification on graphs utilizing graph neural networks.

**Weaknesses:**

1. In the Introduction, the authors assert that "i) To the best of our knowledge, we are the first to learn node embeddings using the abstention-based GAT architecture." This claim seems overstated.

2. In Section 3.1, the authors introduce NodeCwR-Cov and mention that "There are two more fully connected layers after the softmax layer (with 512 nodes and one node) to model the selection function g." The meaning of "having 512 nodes and one node" is unclear in this context.
Additionally, the selection function threshold is set to 0.5, but the rationale behind choosing this value and its impact on the model or performance is not explained.
This threshold serves to filter eligible candidates. It is essential to consider the accuracy of these candidates for each threshold, as they significantly impact the overall performance.

3. The presentation of results in tables and figures is unclear. For instance, in Table 1, the meanings of Cov and LS are not explained.
The experimental analysis lacks depth and clarity.

4. GAT is chosen as the backbone for the proposed model. How does it compare to other graph neural network models?

5. In my opinion, the contribution of this paper appears somewhat limited, and the proposed model seems incremental in its approach.

**Questions:**

Please see the Weaknesses.